# TR-MERGING: TRAINING-FREE ROUTER FOR MODEL MERGING

## ABSTRACT

With the rapid advancement of deep learning, a wide variety of open-source models for different tasks have emerged. However, a single fine-tuned model often fails to meet users' diverse requirements. To address this limitation, model merging has been proposed as an effective approach to integrate the capabilities of existing models into a unified one. Among existing approaches, router-based methods have become representative baselines due to their strong performance; however, their reliance on a trainable router compromises the appealing advantage of traditional model mergingbeing completely training-free. In this paper, we propose a training-free router from a similarity-based perspective. Our method achieves performance on par with router-based approaches while eliminating the need for any additional training. We demonstrate the effectiveness of TR-Merging across multiple tasks in both computer vision (CV) and natural language processing (NLP), and demonstrate its flexibility in adapting to diverse requirements.

## 1 INTRODUCTION

With the rapid progress of deep learning, a wide range of model architectures and training strategies have been introduced, substantially enhancing the capabilities of pre-trained models and positioning them as a cornerstone of modern machine learning. Fine-tuning pre-trained models for downstream tasks has become a prevailing paradigm in both NLP (Devlin et al., 2019; Fan et al., 2024; Lu et al., 2024; Su et al., 2024a;b;c; Sun et al., 2023; Touvron et al., 2023) and CV (Paul & Chen, 2021; Dodge et al., 2020; Dosovitskiy et al., 2021; Ye et al., 2023)domains, often yielding superior performance even with limited labeled data. The proliferation of open-source repositories, such as Huggingface (Wolf et al., 2020), torchvision (Albardi et al., 2021) and ModelScope (Wang et al., 2023), has further accelerated this trend, resulting in an exponential increase in the number of pre-trained and fine-tuned checkpoints. However, maintaining separate models for diverse tasks leads to prohibitive storage and deployment overheads. Multi-task learning (MTL) offers a partial solution by jointly training models across multiple datasets, but it is hindered by substantial computational overhead and constraints on data availability due to privacy concerns. More recently, model merging has emerged as a promising alternative, integrating models through weight combination rather than additional training, thereby addressing these limitations and demonstrating both theoretical significance and broad practical potential.

A straightforward baseline for model merging is direct weight averaging, yet, this often results in substantial performance degradation. To mitigate this issue, several strategies have been proposed, which can be broadly classified into four categories.

- Weighted parameter averaging methods, such as Fisher-Merging (Matena & Raffel) and Reg-Mean (Jin et al., 2022), which employ pre-computed Fisher information or inner product matrices to determine adaptive averaging coefficients.
- Task vector-based methods, including Task Arithmetic (Jiang et al., 2024; Ortiz-Jimenez et al., 2023a; Tang et al., 2023; Yang et al., 2023; Ortiz-Jimenez et al., 2023b; Tang et al., 2024), and AdaMerging (Yang et al., 2023). These approaches merge task vectors rather than raw model parameters, where Ties-Merging explicitly addresses interference, while AdaMerging adaptively adjusts merging coefficients.
- Preprocessing techniques, exemplified by DARE (Yu et al., 2024), which reduces interference by discarding a large portion of task vector elements and rescaling the remaining ones.

- Router-based methods, such as Twin-Merging (Yu et al., 2024) and Free-Merging (Xu et al., 2024), which dynamically route inputs to specialized experts.

Among these categories, router-based approaches generally achieve superior performance. Nonetheless, they introduce an additional training component-Router, which compromises the training-free nature traditionally associated with model merging, thereby incurring extra computational, data, and labor costs in deployment.

Therefore, motivated by distance metric theory, we propose a training-free router as an alternative to conventional training-based routers.Prior approaches typically rely on classifiers trained on domain-specific data of each expert model, which, during inference, route an input to the top-k experts with the highest specialization scores. Although they achieve strong performance, such routers require additional training. In contrast, our method eliminates this requirement by embedding both the input and the domain data of each expert model into a shared representation space and computing their pairwise distances. The input is then assigned to the top-k closest domains, and the corresponding expert models are selected as the most relevant experts for inference.

We empirically demonstrate the effectiveness of TR-Merging, as summarized in Figure 1 First, we merge five Qwen2.5-0.5B-Instruct (Team, 2024) models to validate its performance in the NLP domain. Next, we merge ten ViT-Base-Patch16-224 models to confirm its efficacy in computer vision tasks. To assess scalability, we merge five Qwen2.5-7B-Instruct (Team, 2024) models, illustrating that the approach can be applied to larger models. Furthermore, by merging a classification model with a mathematical reasoning model, we show that TR-merging supports cross-domain and cross-task integration. Finally, evaluation on the out-of-domain MMLU (Hendrycks et al., 2021) benchmark demonstrates that the merged models exhibit strong generalization and robustness.

Our contributions are threefold. First, we introduce TR-Merging, a novel model merging method that leverages a training-free router to integrate task-specific models into a unified model without necessitating additional training. Second, we demonstrate the effectiveness of TR-Merging across a comprehensive suite of both established and newly proposed benchmarks, spanning CV, NLP, PEFT (Hu et al., 2022; Liu et al., 2022; Pei & Wang, 2023; Pei et al., 2024), and multimodal tasks. Finally, we provide a theoretical analysis establishing the optimality of the distance metric algorithm employed by the training-free router within TR-Merging.

## 2 RELATED WORK

This section reviews research on model merging, with a focus on multi-task learning and mixture-of-experts (MoE) frameworks. The goal of model merging is to consolidate multiple task-specific fine-tuned models into a single unified multitask model without necessitating additional training. Initial strategies, such as FisherMerging and RegMean, rely on straightforward weight averaging, yet they often require extra data and substantial computational resources. Another line of work explores interpolation between models within a shared low-loss region, grounded in the concept of linear mode connectivity (LMC) (Draxler et al., 2018; Frankle et al., 2020; Garipov et al., 2018). To facilitate effective parameter alignment for interpolation, methods like *weight matching* and *optimal transport* have been introduced, although recent evidence indicates that LMC assumptions may not consistently apply to fine-tuned models. Task-Arithmetic generalizes simple averaging by performing more flexible arithmetic operations in parameter space, enabling finer control over model behavior. Nevertheless, interference among tasks remains a significant challenge. To address this, techniques such as Ties-Merging, AdaMerging, and DARE (Yu et al., 2024) have been developed to mitigate task conflicts by identifying redundant parameters, learning optimal merging coefficients, and reducing parameter density. Twin-Merging (Yu et al., 2024) further proposes a modular knowledge composition mechanism that dynamically integrates knowledge modules based on their relevance. While some approaches exploit task identities at inference time to enhance merging performance, such assumptions are often unrealistic in practical scenarios where task distributions are unknown or variable.

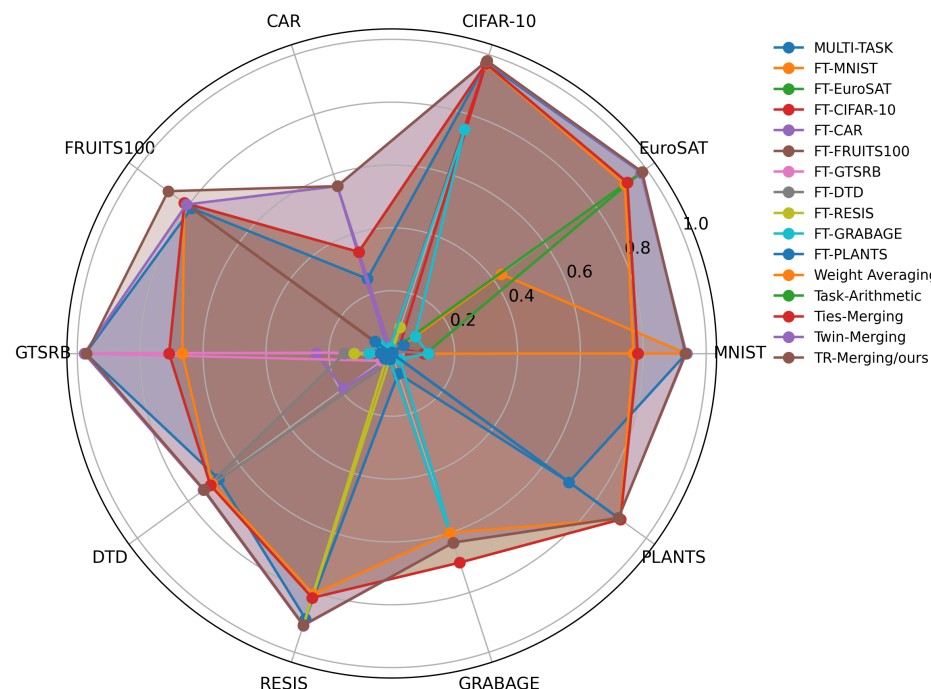

Figure 1: Comparison accuracy (%) with Existing Methods Demonstrating the Superior Effectiveness of Our Approach.

## 3 METHOD

This chapter outlines the complete workflow of our proposed method. Following the preliminary setup, the approach is structured into two main stages(as illustrated in Figure 2): In Section 3.1, a training-free Router is utilized to generate embeddings, from which similarity scores are computed to derive the weighting coefficients of experts. Section 3.2 then leverages these coefficients to perform model merging. To further substantiate the methodological soundness, Section 3.3 presents the corresponding theoretical foundations.

**Task denotes.** Given $N$ tasks $[T_1, \ldots, T_N]$, the goal of model merging is to obtain a single model suitable for all tasks by using the models $[\theta_1, \ldots, \theta_N]$ fine-tuned from the same pretrained model $\boldsymbol{\theta}_{pre}$. Existing methods focus on merging these models into a unified model $\boldsymbol{\theta}_m$. It is important to note that we adopt LoRA as an efficient fine-tuning method, which is more compatible with our approach. Compared with full-parameter fine-tuning, LoRA can reduce memory consumption during inference and improve inference speed.

### 3.1 EXPERT WEIGHT DERIVATION VIA TRAINING-FREE ROUTER

Task arithmetic has been widely recognized as a fundamental principle in model merging, and it can be formally expressed as:

$$\boldsymbol{\theta}_m = \boldsymbol{\theta}_{pre} + \sum \lambda(\boldsymbol{\theta}_i - \boldsymbol{\theta}_{pre}), \tag{1}$$

Here, $\theta_i - \theta_{pre}$ captures the domain-specific knowledge of each expert, denoted as $\Delta_i$, while $\lambda$ represents the weighting factor quantifying each expert's contribution to the merged model. During inference, however, the influence of individual experts is task-dependent. To accommodate this, Twin-Merging employs a Router that is trained to adaptively assign expert weights conditioned on the input. Specifically, given an input $\mathbf{X}$, the Router produces contribution coefficients $\alpha_i$, yielding the merged model as:

$$\boldsymbol{\alpha} = \text{Router}(\mathbf{X}), \quad \boldsymbol{\alpha} = (\alpha_1, \ldots, \alpha_N) \tag{2}$$

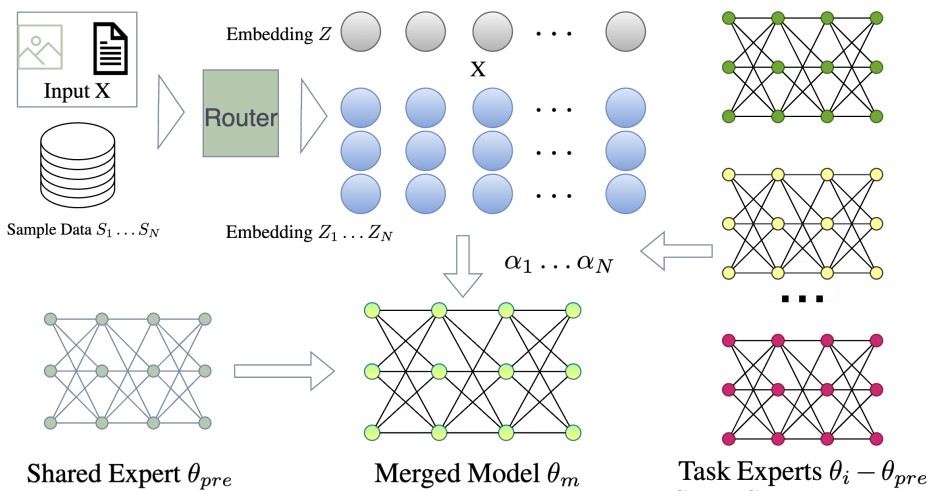

Figure 2: Overview of the proposed method, illustrating its main workflow.

$$\boldsymbol{\theta}_m = \boldsymbol{\theta}_{pre} + \sum \alpha_i(\boldsymbol{\theta}_i - \boldsymbol{\theta}_{pre}), \tag{3}$$

This method indeed achieves excellent performance; however, the introduction of the additional trained component, the Router, breaks the training-free advantage of model merging and further increases deployment costs. Motivated by similarity-based approaches, we propose a **training-free expert routing mechanism based on semantic space correlation**. The central idea is to directly explore the correlation between the semantic space of the model input and the semantic space of the expert models' training data, thereby enabling dynamic expert selection and weighting.

Formally, let the task set be $[T_1, \ldots, T_N]$. For each task $T_i$, we sample a subset of training data $\mathcal{S}_i = \{x_i^1, \ldots, x_i^M\}$ to represent the semantic distribution of the task. We introduce a pretrained embedding model $\mathcal{R}(\cdot)$ as the Router to encode both the input sample $\mathbf{x}$ and the sampled data from each task:

$$\mathbf{z} = \mathcal{R}(\mathbf{x}), \quad \mathbf{z}_i^j = \mathcal{R}(x_i^j), \quad j = 1, \ldots, M. \tag{4}$$

We then compute the cosine similarity between the input representation $\mathbf{z}$ and the set of sampled representations from task $T_i$:

$$s_i(\mathbf{x}) = \frac{1}{M} \sum_{j=1}^{M} \frac{\langle \mathbf{z}, \mathbf{z}_i^j \rangle}{\|\mathbf{z}\| \cdot \|\mathbf{z}_i^j\|}. \tag{5}$$

The score $s_i(\mathbf{x})$ characterizes the semantic proximity between the input $\mathbf{x}$ and task $T_i$. We further normalize the relevance scores across all tasks via a softmax transformation:

$$\alpha_i = \frac{\exp(s_i(\mathbf{x}))}{\sum_{k=1}^{N} \exp(s_k(\mathbf{x}))}, \quad i = 1, \ldots, N. \tag{6}$$

## 3.2 DYNAMIC MODEL MERGING

In the previous step, we obtained the task relevance weights $\{\alpha_i\}_{i=1}^N$. However, directly using this distribution may lead to insufficient contrast among experts, thereby reducing the discriminative power of expert selection. To address this, we introduce a temperature scaling coefficient $\tau > 0$ to sharpen the distribution, amplifying the weights of highly relevant experts while suppressing those of less relevant ones. Specifically, we first scale the raw weights as:

$$\tilde{\alpha}_i = \frac{\alpha_i}{\tau}, \quad i = 1, \ldots, N. \tag{7}$$

Then, we apply a softmax normalization over the scaled values to obtain the temperature-adjusted weights:

$$\hat{\alpha}_i = \frac{\exp(\tilde{\alpha}_i)}{\sum_{k=1}^N \exp(\tilde{\alpha}_k)}, \quad i = 1, \dots, N. \tag{8}$$

This transformation preserves numerical stability while enhancing the distinction between the input and its most relevant tasks, making the expert routing more focused on the truly informative experts.

On this basis, we select the top-$K$ experts with the highest weights, denoted by $\mathcal{I}_K(\mathbf{x})$, and use them for the final model merging:

$$\boldsymbol{\theta}_m = \boldsymbol{\theta}_{pre} + \sum_{i \in \mathcal{I}_K(\mathbf{x})} \hat{\alpha}_i \cdot \left(\boldsymbol{\theta}_i - \boldsymbol{\theta}_{pre}\right). \tag{9}$$

By introducing temperature scaling and softmax normalization, our method adaptively amplifies the influence of salient experts while suppressing the impact of redundant ones, thereby enhancing both the robustness and representational capacity of the merged model. Crucially, this design preserves task-level semantic alignment without incurring the additional training cost of a Router, thus retaining the inherent "training-free" advantage of model merging. Consequently, the approach enables robust and efficient expert selection as well as effective parameter synthesis.

### 3.3 THEORETICAL SUPPORT FROM GAUSSIAN SIMILARITY MODELING

To provide a theoretical understanding of the proposed training-free expert routing mechanism, we analyze the expert selection process from the perspective of Gaussian similarity modeling. We assume that the embeddings of input data $\mathbf{x}$ and the sampled task-specific representations $\mathbf{z}_i^j$ are drawn from multivariate Gaussian distributions:

$$\mathbf{z} \sim \mathcal{N}(\boldsymbol{\mu}_x, \Sigma_x), \quad \mathbf{z}_i^j \sim \mathcal{N}(\boldsymbol{\mu}_i, \Sigma_i), \quad j = 1, \dots, M, \tag{10}$$

where $\boldsymbol{\mu}_x$ and $\boldsymbol{\mu}_i$ denote the mean embedding vectors of the input and the $i$-th task, and $\Sigma_x$ and $\Sigma_i$ denote their corresponding covariance matrices.

We now analyze the statistical properties of the similarity scores under Gaussian assumptions.

**Expected Similarity.** The cosine similarity $s_i(\mathbf{x})$ between the input and the sampled representations from task $T_i$ can be approximated by the inner product of normalized Gaussian means:

$$\mathbb{E}[s_i(\mathbf{x})] \approx \frac{\boldsymbol{\mu}_x^\top \boldsymbol{\mu}_i}{\|\boldsymbol{\mu}_x\| \cdot \|\boldsymbol{\mu}_i\|}. \tag{11}$$

This approximation holds under the assumption that the embeddings within each task cluster are tightly concentrated around the mean, i.e., $\text{Tr}(\Sigma_i) \ll \|\boldsymbol{\mu}_i\|^2$.

**Concentration Inequality.** By applying Hoeffding's inequality for bounded random variables, or equivalently, using standard concentration results for Gaussian variables, the empirical similarity computed over $M$ samples concentrates around its expectation with high probability:

$$\mathbb{P}\Big(\big|s_i(\mathbf{x}) - \mathbb{E}[s_i(\mathbf{x})]\big| \geq \epsilon\Big) \leq 2\exp\Big(-\frac{M\epsilon^2}{2\sigma^2}\Big), \tag{12}$$

where $\sigma^2$ denotes the variance of the pairwise cosine similarities. This result guarantees that with a sufficiently large $M$, the estimated $\alpha_i$ reliably reflects the true semantic proximity between the input and task $T_i$.

**Temperature Scaling Interpretation.** Introducing a temperature $\tau > 0$ in the softmax is equivalent to sharpening the probability distribution over experts, which increases the likelihood of selecting experts whose embeddings are closest to the input in the Gaussian sense:

$$\hat{\alpha}_i = \frac{\exp(s_i(\mathbf{x})/\tau)}{\sum_{k=1}^N \exp(s_k(\mathbf{x})/\tau)}. \tag{13}$$

From an information-theoretic perspective, this transformation increases the KL-divergence between the selected top-$K$ experts and the uniform distribution over all experts, thus improving the discriminative power of expert selection.

**Top-$K$ Expert Recovery.** Assuming that the task means $\boldsymbol{\mu}_i$ are sufficiently separated in the embedding space, i.e.,

$$\min_{i \neq j} \frac{\|\boldsymbol{\mu}_i - \boldsymbol{\mu}_j\|}{\max\{\|\boldsymbol{\mu}_i\|, \|\boldsymbol{\mu}_j\|\}} \geq \delta, \tag{14}$$

then with high probability, the top-$K$ experts selected via $\hat{\alpha}_i$ correspond to the $K$ most semantically relevant tasks. This guarantees that the final merged model:

$$\boldsymbol{\theta}_m = \boldsymbol{\theta}_{pre} + \sum_{i \in \mathcal{I}_K(\mathbf{x})} \hat{\alpha}_i(\boldsymbol{\theta}_i - \boldsymbol{\theta}_{pre}) \tag{15}$$

incorporates the most relevant domain knowledge while avoiding interference from irrelevant experts.

**Robustness Implication.** This robustness guarantee ensures that the merged model remains stable under small input perturbations, which is essential for reliable deployment in practical applications. Since the Gaussian assumption implies bounded variance of embeddings, the combination of temperature scaling and top-$K$ selection ensures that the merged parameters $\boldsymbol{\theta}_m$ remain stable under small perturbations of the input $\mathbf{x}$. Formally, for $\mathbf{x}' = \mathbf{x} + \Delta\mathbf{x}$ with $\|\Delta\mathbf{x}\| \leq \epsilon$, we have

$$\|\boldsymbol{\theta}_m(\mathbf{x}') - \boldsymbol{\theta}_m(\mathbf{x})\| \leq C\epsilon, \tag{16}$$

where $C$ depends on the sensitivity of the embeddings and the expert weights, ensuring smooth and robust expert routing.

Overall, this Gaussian-based theoretical analysis justifies that the proposed training-free routing mechanism can reliably select relevant experts, amplify their contributions, and yield a robust merged model without additional training overhead.

## 4 EXPERIMENT

In this section, we conduct a thorough evaluation of the proposed TR-merging framework across diverse experimental conditions, covering cross-task, cross-domain, heterogeneous training configurations, as well as domain-shift settings. To assess its effectiveness, we benchmark our approach against three widely studied baselines: Weight Averaging, Task Arithmetic, Ties-Merging and Twin-Merging. Specifically, Section 4.1 reports the results of TR-merging on both natural language processing benchmarks (Pei et al., 2019) and computer vision datasets, while Section 4.2 investigates its scalability to a larger pool of models and analyzes its ability to generalize across domains and tasks.

### 4.1 COMPARATIVE EVALUATION

**Setup.** We conduct experiments on five NLP datasets: **RTE**, **MNLI**, **QNLI**, **QQP**, **MRPC** (Wang et al., 2018), and ten CV datasets: **MNIST** (LeCun et al., 2010), **EuroSAT** (Helber et al., 2019), **CIFAR-10** (Krizhevsky, 2009), **CarBrands50**, **Fruits100**, **GTSRB** (Stallkamp et al., 2011), **DTD** (Cimpoi et al., 2014), **RESISC** (Cheng et al., 2017), **GRABAGE**, **PLANTS**.

For the NLP experiments, our method along with all baseline approaches is evaluated on the Qwen2.5-0.5B-Instruct model (Yang et al., 2024; Team, 2024). In the computer vision setting, we instead utilize the pretrained ViT-Base-Patch16-224 backbone as the reference architecture. Unless otherwise noted, input images are consistently normalized to a resolution of $224 \times 224$ for both the training and inference phases. To ensure comparability across domains, we employ a unified evaluation protocol: throughput is reported on a single NVIDIA A100 GPU with batch size fixed to 32 under FP32 precision; classification accuracy serves as the principal performance indicator; and efficiency is measured in terms of memory footprint and inference latency. Additional dataset descriptions and implementation specifics can be found in Appendix C.

**Implementation detail** For **NLP** tasks, we adopt the LoRA fine-tuning framework (Hu et al., 2022), using a rank of 8 and a scaling factor of 32. Starting from the pretrained ViT (Dosovitskiy et al., 2020), we fine-tune it on five benchmark datasetsRTE, MNLI, QNLI, QQP, and MRPCresulting in task-specialized variants denoted as FT-RTE, FT-MNLI, FT-QNLI, FT-QQP, and FT-MRPC.

The optimization is performed using AdamW with a learning rate of $1 \times 10^{-4}$. For **CV** tasks, we similarly follow the LoRA fine-tuning protocol but configure a rank of 16 and a scaling factor of 16. The ViT backbone is pretrained and subsequently adapted on MNIST, EuroSAT, CIFAR-10, CarBrands50, and Fruits100, producing FT-MNIST, FT-EuroSAT, FT-CIFAR-10, FT-CAR, and FT-FRUITS100. The AdamW optimizer is employed with a learning rate of $5 \times 10^{-3}$. In **both domains**, LoRA adapters are injected into the MLP layers with a dropout rate of 0.1, while bias parameters remain frozen. Training is stabilized using a warm-up learning rate schedule, and cross-entropy loss is minimized with weight decay set to 0.01. To ensure reproducibility, random seeds are fixed across NumPy, PyTorch, and Python. For merging baselines, we observe that setting the task arithmetic coefficient to 0.3 consistently provides superior results. Inference latency is computed as the average over 100 full-dataset runs to yield stable measurements. All experiments are carried out on a single NVIDIA A100 GPU with CUDA 12.4, cuDNN 9.1.0, and PyTorch 2.1.2.

For **expert routing**, we leverage lightweight embedding encoders to represent inputs and guide expert selection. In the CV setting, we employ CLIP-ViT-B/16, while in NLP we utilize BGE-Small-en-v1.5. These models act as efficient Routers, delivering compact yet informative routing signals for task-aligned expert activation.

Table 1: Comparison of task-specific fine-tuned models, merge baselines, and our method across GLUE tasks.

| MODEL | RTE | MNLI | QNLI | QQP | MRPC | AVG | VRAM | TIME |
|---|---|---|---|---|---|---|---|---|
| **MULTI-TASK** | 77.4% | 81.1% | 83.0% | 78.0% | 76.2% | 79.1% | 2010M | 184s |
| **FT-RTE** | **77.3%** | 52.8% | 56.0% | 61.0% | 58.9% | 61.2% | 2010M | 195s |
| **FT-MNLI** | 71.5% | **82.0%** | 33.4% | 62.0% | 63.3% | 62.4% | 2010M | 195s |
| **FT-QNLI** | 62.5% | 46.8% | **84.0%** | 65.0% | 65.7% | 64.8% | 2010M | 195s |
| **FT-QQP** | 64.3% | 43.2% | 64.4% | **84.8%** | 70.4% | 65.4% | 2010M | 195s |
| **FT-MRPC** | 49.5% | 36.8% | 56.0% | 65.8% | **85.3%** | 58.7% | 2010M | 195s |
| **Weight Averaging** | 68.2% | 39.6% | 67.8% | 64.0% | 57.1% | 59.4% | 2010M | 195s |
| **Task-Arithmetic** | 66.8% | 65.6% | 59.2% | 71.6% | 74.0% | 66.4% | 2010M | 195s |
| **Ties-Merging** | 66.4% | 65.2% | 59.6% | 70.4% | 67.9% | 65.9% | 2010M | 195s |
| **Twin-Merging** | 76.9% | 81.8% | 83.6% | 84.8% | 85.0% | 82.4% | 2010M+($N$-1)*34.2M | 275s |
| **TR-merging/ours** | **78.3%** | **82.2%** | **84.4%** | **85.0%** | **85.1%** | **83.0%** | 2010M+($N$-1)*20M | 249s |

**Results.** As reported in Tables 1 and 2, our approach consistently outperforms representative model merging baselines across both NLP and CV benchmarks. Remarkably, it even achieves better single-task performance than individually fine-tuned models, while incurring only negligible additional storage and runtime overhead. These results highlight the capability of our method to effectively mitigate long-standing challenges in model merging, including parameter conflicts and task interference. Unlike many prior approaches that struggle to scale under such conditions, our method demonstrates stronger robustness and scalability, underscoring its potential for broader multi-domain applications.

## 4.2 COMPARATIVE ANALYSIS

**Large-Model Scalability and Out-of-Domain Generalizability.** Scaling model merging to larger architectures has emerged as a critical research problem. To assess the scalability of our method, we extended the experiments to Qwen2.5-7B-Instruct under the same training configurations as Qwen2.5-0.5B-Instruct. As shown in Table 4, our approach continues to deliver strong results, even surpassing fine-tuned baselines, demonstrating that its effectiveness is not limited by model size. Another long-standing challenge in model merging lies in out-of-domain (OOD) generalization, as merged models typically integrate only a restricted set of expert competencies. Consequently, performance degradation is often observed when encountering tasks beyond the training domains. Nevertheless, as reported in Table 3 (with detailed results in Table 6), our merged Qwen2.5-7B-Instruct models trained on RTE, MNLI, QNLI, QQP, and MRPC outperform both Weight Averaging and

Table 2: Comparison of task-specific fine-tuned models, merge baselines, and our method across CV tasks.

| MODEL | MNIST | EuroSAT | CIFAR-10 | CAR | FRUITS100 | GTSRB | DTD | RESIS | GRABAGE | PLANTS | AVG | VRAM | TIME |
|---|---|---|---|---|---|---|---|---|---|---|---|---|---|
| **MULTI-TASK** | 93.5% | 98.1% | 97.1% | 46.0% | 81.6% | 98.2% | 69.7% | 91.2% | 66.7% | 74.1% | 75.6% | 805M | 158s |
| **FT-MNIST** | **93.6%** | 4.3% | 0.1% | 0.0% | 0.1% | 0.6% | 0.4% | 0.2% | 0.0% | 0.1% | 9.9% | 805M | 158s |
| **FT-EuroSAT** | 10.9% | **98.0%** | 1.1% | 0.0% | 0.4% | 4.0% | 1.2% | 2.7% | 0.0% | 1.1% | 11.9% | 805M | 158s |
| **FT-CIFAR-10** | 10.2% | 2.8% | **97.8%** | 2.0% | 2.9% | 4.1% | 0.7% | 2.9% | 0.0% | 0.6% | 12.4% | 805M | 158s |
| **FT-CAR** | 0.0% | 1.0% | 0.1% | **56.0%** | 1.5% | 2.4% | 1.9% | 1.6% | 0.0% | 0.6% | 6.5% | 805M | 158s |
| **FT-FRUITS100** | 1.1% | 1.9% | 0.1% | 0.0% | **80.7%** | 0.4% | 0.0% | 0.2% | 0.0% | 1.9% | 8.6% | 805M | 158s |
| **FT-GTSRB** | 2.3% | 0.1% | 0.6% | 0.0% | 0.5% | **97.8%** | 0.4% | 0.8% | 0.0% | 0.1% | 10.3% | 805M | 158s |
| **FT-DTD** | 11.8% | 1.2% | 1.8% | 1.0% | 0.1% | 1.5% | **73.8%** | 3.2% | 0.0% | 2.6% | 9.7% | 805M | 158s |
| **FT-RESIS** | 0.0% | 0.0% | 8.7% | 1.0% | 0.4% | 1.2% | 1.9% | **91.1%** | 0.0% | 0.0% | 10.4% | 805M | 158s |
| **FT-GRABAGE** | 11.6% | 9.2% | 7.5% | 2.0% | 5.0% | 7.3% | 2.2% | 2.2% | **63.3%** | 3.3% | 11.4% | 805M | 158s |
| **FT-PLANTS** | 0.2% | 4.6% | 0.4% | 0.0% | 6.5% | 3.5% | 2.8% | 1.9% | 0.0% | **89.1%** | 10.9% | 805M | 158s |
| **Weight Averaging** | 76.9% | 91.6% | 96.8% | 34.0% | 81.6% | 66.6% | 70.6% | 80.6% | 60.0% | 89.8% | 74.8% | 805M | 158s |
| **Task-Arithmetic** | 78.2% | 92.6% | 97.0% | 34.0% | 81.5% | 70.8% | 71.3% | 81.8% | 70.0% | 89.9% | 76.7% | 805M | 158s |
| **Ties-Merging** | 78.2% | 92.6% | 97.0% | 34.0% | 81.5% | 70.8% | 71.3% | 81.8% | 70.0% | 89.9% | 76.7% | 805M | 158s |
| **Twin-Merging** | 93.6% | 98.0% | 97.8% | 56.0% | 80.7% | 97.8% | 74.0% | 91.1% | 63.3% | 89.1% | 84.1% | 805M+($N$-1)*7M | 223s |
| **TR-merging/ours** | **93.4%** | **98.4%** | **98.0%** | **56.0%** | **87.9%** | **97.1%** | **73.8%** | **90.9%** | **63.3%** | **89.1%** | **84.8%** | 805M+($N$-1)*5M | 184s |

Task Arithmetic, and achieve results that are nearly comparable to general-purpose models. This unexpected finding offers compelling evidence that TR-merging preserves strong OOD generalization ability.

Table 3: Average Performance on Out-of-Domain MMLU Benchmark Tasks

| MMLU Task | Qwen2.5-7B-Instruct | Weight Averaging | Task-Arithmetic | TR-merging/ours |
|---|---|---|---|---|
| **Avg** | 67.9% | 67.3% | 63.6% | **67.5%** |

Table 4: Task-Level Performance of Our Method on Qwen2.5-7B-Instruct Models

| MODEL | MNLI | MRPC | QNLI | QQP | RTE | AVG |
|---|---|---|---|---|---|---|
| **Pretrain** | 53.8% | 55.0% | 48.0% | 51.6% | 50.4% | 51.8% |
| **MULTI-TASK** | 84.8% | 86.4% | 86.4% | 88.8% | 86.6% | 86.6% |
| **Finetune** | 89.2% | 89.0% | 92.0% | 86.0% | 91.7% | 89.6% |
| **TR-merging/ours** | **89.2%** | **89.3%** | **92.0%** | **86.2%** | **91.7%** | **89.7%** |

**Task Diversity and Span.** As shown in Tables 1 2, our method demonstrates robust performance across both computer vision (CV) and natural language processing (NLP) tasks, covering a diverse set of classification scenarios such as 2 (GARBAGE), 3 (RTE), 10 (EuroSAT), 30 (PLANTS), 43 (GTSRB), 45 (RESISC), 47 (DTD), 50 (CarBrands50), and 100 (Fruits100).

Table 5: Performance Across Different Task Types

| Method | Classification | Generation |
|---|---|---|
| **Funetune** | 77.3% | 49.5% |
| **TR-merging/ours** | **77.8%** | **49.9%** |

These tasks span a broad range of semantic domains, including digits, remote sensing, general objects, automobiles, fruits, plants, waste, traffic signs and various other domains, high lighting the method's ability to generalize across distinct knowledge areas. Furthermore, as illustrated in Table 5, our approach supports the integration of both generative and discriminative tasks within a unified framework, marking a pioneering effort in

this direction. Notably, it even outperforms fine-tuned models that are often regarded as the theoretical upper bound. These results collectively underscore the exceptional scalability of our method across modalities, knowledge domains, and task types.

**Scalability to Large-Scale Tasks.** As shown in Tables 1 and 2, our approach achieves consistently strong results across both computer vision (CV) and natural language processing (NLP) tasks, spanning diverse classification settings such as 2-way (GARBAGE), 3-way (RTE), 10-way (EuroSAT), 30-way (PLANTS), 43-way (GTSRB), 45-way (RESISC), 47-way (DTD), 50-way (CarBrands50), and 100-way (Fruits100). These benchmarks encompass a wide spectrum of semantic domains, ranging from digits and remote sensing to general objects, vehicles, fruits, plants, waste management, and traffic sign recognition, thereby underscoring the models capacity to generalize across heterogeneous knowledge sources. Moreover, as reported in Table 5, our method is capable of integrating both discriminative and generative tasks within a unified frameworkrepresenting one of the first attempts in this direction. Remarkably, it even surpasses fine-tuned baselines, which are typically considered the theoretical upper bound of task-specific performance. Taken together, these findings highlight the strong scalability of our approach across modalities, domains, and task paradigms.

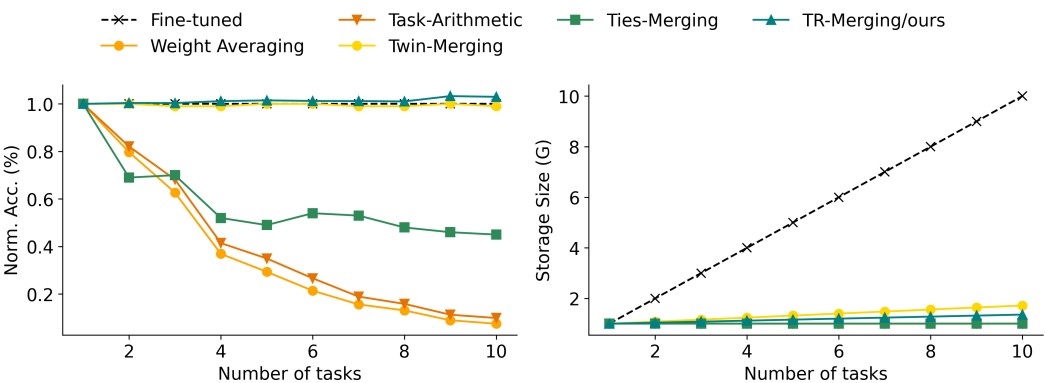

Figure 3: Scalability analysis of model accuracy and storage footprint as the number of tasks increases.

## 5 CONCLUSION

In this work, we have introduced **TR-Merging**, a novel model merging framework that leverages a *training-free router* to integrate task-specific models into a single unified model without incurring additional training costs. By exploiting semantic similarity between input data and task domains, our method adaptively selects and weights the most relevant experts, preserving task-specific knowledge while mitigating interference from irrelevant tasks. We further incorporated temperature scaling and top-$K$ selection to enhance the discriminative power and robustness of expert routing. Extensive experiments across NLP and CV tasks, as well as cross-domain and cross-task scenarios, demonstrate that TR-Merging consistently outperforms existing model merging baselines, including weight averaging, task arithmetic, and router-based methods, both in terms of accuracy and computational efficiency. Theoretical analysis based on Gaussian similarity modeling provides formal support for the effectiveness and stability of our training-free routing mechanism. Overall, TR-Merging offers a practical, scalable, and efficient solution for integrating multiple task-specific models, preserving the advantages of traditional training-free merging while achieving performance on par with more complex router-based approaches. We believe that this work paves the way for broader adoption of training-free expert routing in multi-task and multi-domain model deployment, enabling flexible adaptation to diverse user requirements without additional computational burden.

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

# A  MORE RELATIVE RESEARCH

**Averaging.**  Parameter averaging is a well-established technique in federated learning Recent applications have extended its utility to model merging for enhancing robustness against out-of-distribution data, refining pre-trained models, developing multimodal architectures, and creating multitask models by combining model capabilities. Parameter averaging is performed by computing the mean of all expert model weights, without relying on a base model. Formally, this can be expressed as:

$$\mathcal{M}(\{\theta_i\}_{i=1}^N, \theta_{\text{base}}) = \frac{1}{N} \sum_{i=1}^{N} \theta_i.$$

**Fisher Merging.**  The method assesses the significance of each parameter when merging models for task $t$ by computing the Fisher information matrix. The matrix is given by the following formula:

$$\hat{F}_t = \mathbb{E}_{x \sim D_t} \mathbb{E}_{y \sim p_{\theta_t}(y|x)} \nabla_{\theta_t} (\log p_{\theta_t}(y|x_t))^2,$$

where the model merging is guided by this significance measure.

**RegMean.**  The method imposes a constraint on the model merging process by minimizing the $L_2$ distance between the activations of the merged model and those of the individual models. It achieves this by computing the least-squares solution given by

$$\theta_m = \left( \sum_{t=1}^{n} X_t^T X_t \right)^{-1} \sum_{t=1}^{n} (X_t^T X_t \theta_t),$$

where $X_t$ represents the input activation of the corresponding layer.

**Task Arithmetic.**  Task Arithmetic introduces a novel concept of *task vectors* for model merging. For a given task $\mathbf{t}_i$, the corresponding task vector is defined as $\boldsymbol{\tau}_i = \boldsymbol{\theta}_i - \boldsymbol{\theta}_{\text{base}}$, which captures task-specific knowledge by quantifying the difference between the fine-tuned expert parameters $\boldsymbol{\theta}_i$ and the original base model parameters $\boldsymbol{\theta}_{\text{base}}$. A scaling hyperparameter $\lambda$ governs the contribution of the aggregated task-specific knowledge to the final model. The merged model is constructed by linearly combining the base model parameters with a scaled sum of all task vectors. Formally, task arithmetic is defined as:

$$\mathcal{M}(\{\boldsymbol{\theta}_i\}_{i=1}^N, \boldsymbol{\theta}_{\text{base}}; \lambda) = \boldsymbol{\theta}_{\text{base}} + \lambda \cdot \sum_{i=1}^{N} (\boldsymbol{\theta}_i - \boldsymbol{\theta}_{\text{base}}).$$

**AdaMerging.**  The method automatically learns a merging coefficient for each layer of each task vector in Task Arithmetic.

**Ties-Merging.**  TIES-Merging identifies two major challenges in model merging: Fine-tuned expert models often accumulate substantial noise in their parameters;Different experts may attempt to update the same parameter in conflicting directions, causing interference between models. To address these issues, TIES-Merging introduces a three-step procedure: First, removing redundant parameters. Second, resolving sign conflicts. Third, aggregating only the non-conflicting parameters. Specifically, for each task $i$, parameters in the task vector with small magnitudes are zeroed out to produce the trimmed task vector $\hat{\tau}_i$. Then, for each parameter $p$, the aggregate sign $\gamma_m^p$ is determined by the sign of the sum of corresponding entries across all trimmed task vectors:

$$\gamma_m^p = \text{sgn} \left( \sum_{i=1}^{N} \hat{\tau}_i^p \right).$$

Next, only those models whose trimmed task vector entries match the aggregate sign are included in the merging process. That is, the index set of participating models is defined as $\mathcal{A}^p = \{i \in [N] \mid \text{sgn}(\hat{\tau}_i^p) = \gamma_m^p\}$.

Finally, the merged task vector is computed by averaging over the selected models, scaled by a hyperparameter $\lambda$, and added back to the base model parameters:

$$\boldsymbol{\theta}_m^p = \boldsymbol{\theta}_{\text{base}}^p + \lambda \cdot \frac{1}{|\mathcal{A}^p|} \sum_{i \in \mathcal{A}^p} \hat{\tau}_i^p.$$

**Dare Merging.** The method effectively reduces parameter redundancy by setting the majority of delta parameters to zero and rescaling the remaining parameters. This is achieved through the transformation given by

$$\theta' = \frac{\theta}{1 - p},$$

where $p$ represents the proportion of delta parameters that are discarded.

**Twin-Merging.** The method that encompasses two principal stages: modularizing knowledge into shared and exclusive components, with compression to reduce redundancy and enhance efficiency; dynamically merging shared and task specific knowledge based on the input. This approach narrows the performance gap between merged and fine-tuned models and improves adaptability to heterogeneous data.

$$\theta^* = \theta_s + \sum_{t=1}^{T} w_t * \text{SVD}_r(\theta_t - \theta_s)$$

where $\theta_s$ represents the parameter set of the shared expert, which is common across all tasks. The term $\theta_t - \theta_s$ denotes the task expert, capturing the task-specific adjustments to the shared expert parameters. The operation $\text{SVD}_r$ refers to the singular value decomposition applied with a rank constraint $r$, which serves to sparsify the task expert parameters, retaining only the most significant variations.

# B   THEORETICAL ANALYSIS OF TRAINING-FREE EXPERT ROUTING

To provide a rigorous understanding of the proposed training-free expert routing mechanism, we formalize the analysis using Gaussian similarity modeling.

**Lemma B.1** (Expected Similarity). *Assume that the input embedding $\mathbf{z}$ and task-specific embeddings $\mathbf{z}_i^j$ are drawn from multivariate Gaussian distributions:*

$$\mathbf{z} \sim \mathcal{N}(\mu_x, \Sigma_x), \quad \mathbf{z}_i^j \sim \mathcal{N}(\mu_i, \Sigma_i), \quad j = 1, \dots, M. \tag{17}$$

*If the embeddings within each task cluster are concentrated around their mean ($\text{Tr}(\Sigma_i) \ll \|\mu_i\|^2$), then the expected cosine similarity between the input and the $i$-th task is approximated as*

$$\mathbb{E}[s_i(\mathbf{x})] \approx \frac{\mu_x^\top \mu_i}{\|\mu_x\| \, \|\mu_i\|}. \tag{18}$$

*Proof.* Under the Gaussian assumption, the embeddings $\mathbf{z}_i^j$ concentrate near $\mu_i$. Therefore, the cosine similarity between $\mathbf{z}$ and $\mathbf{z}_i^j$ is dominated by the inner product of the means. By linearity of expectation over $M$ samples, the result follows. $\square$

**Lemma B.2** (Concentration of Empirical Similarity). *Let $s_i(\mathbf{x})$ be the empirical cosine similarity computed over $M$ samples from task $T_i$. Then, with high probability,*

$$\mathbb{P}\Big(\big|s_i(\mathbf{x}) - \mathbb{E}[s_i(\mathbf{x})]\big| \geq \epsilon\Big) \leq 2 \exp\Big(-\frac{M\epsilon^2}{2\sigma^2}\Big), \tag{19}$$

*where $\sigma^2$ denotes the variance of pairwise cosine similarities.*

*Proof.* This follows from Hoeffding's inequality for bounded random variables or standard concentration inequalities for Gaussian variables. As $M$ increases, the empirical average converges to its expectation with high probability. $\square$

**Proposition B.3** (Effect of Temperature Scaling). *Introducing a temperature $\tau > 0$ in the softmax distribution sharpens the selection probability of experts:*

$$\hat{\alpha}_i = \frac{\exp(s_i(\mathbf{x})/\tau)}{\sum_{k=1}^{N} \exp(s_k(\mathbf{x})/\tau)}. \tag{20}$$

*Proof.* A smaller $\tau$ increases the difference between the highest and lowest similarity scores in the softmax, effectively amplifying the contribution of highly relevant experts. This can be interpreted as increasing the KL-divergence between the top-$K$ selected experts and the uniform distribution over all experts, thereby improving discriminative power. □

**Theorem B.4** (Top-$K$ Expert Recovery). *Assume task means $\mu_i$ are sufficiently separated:*

$$\min_{i \neq j} \frac{\|\mu_i - \mu_j\|}{\max\{\|\mu_i\|, \|\mu_j\|\}} \geq \delta. \tag{21}$$

*Then, with high probability, the top-$K$ experts selected via $\hat{\alpha}_i$ correspond to the $K$ most semantically relevant tasks. Consequently, the merged model*

$$\theta_m = \theta_{\text{pre}} + \sum_{i \in \mathcal{I}_K(\mathbf{x})} \hat{\alpha}_i(\theta_i - \theta_{\text{pre}}) \tag{22}$$

*integrates the most relevant domain knowledge while minimizing interference from irrelevant experts.*

*Proof.* Given the sufficient separation between task means and the concentration of embeddings (Lemmas 1 and 2), the empirical similarity scores reliably reflect the true semantic relevance. The top-$K$ selection over $\hat{\alpha}_i$ therefore identifies the $K$ tasks closest to the input in semantic space with high probability. The merged model then combines the most relevant updates while excluding irrelevant ones. □

**Corollary B.5** (Robustness under Input Perturbations). *Assuming the embedding function is Lipschitz continuous, small perturbations $\Delta\mathbf{x}$ in the input yield bounded changes in the merged parameters:*

$$\|\theta_m(\mathbf{x} + \Delta\mathbf{x}) - \theta_m(\mathbf{x})\| \leq C\|\Delta\mathbf{x}\|, \tag{23}$$

*where $C$ depends on the sensitivity of embeddings and expert weights.*

*Proof.* By the Lipschitz continuity of embeddings and the bounded effect of temperature-scaled softmax weights, any small perturbation in the input results in a proportionally bounded change in $\theta_m$. This ensures smooth and robust expert routing. □

**Discussion.**

- **Sample Complexity:** Accurate top-$K$ expert recovery requires $M = O(\delta^{-2}\log(N/\epsilon))$, implying that well-separated embeddings enable reliable routing with few samples.

- **Temperature Selection:** Smaller $\tau$ sharpens expert selection, while larger $\tau$ improves robustness against noise.

- **Low-rank LoRA Updates:** If $\theta_i - \theta_{\text{pre}}$ are low-rank (as in LoRA), merged parameters inherit additional stability, tightening perturbation bounds.

- **Scalability:** Approximate nearest-neighbor search of task centroids allows scaling to large expert pools without significantly affecting selection accuracy.

Overall, this Gaussian-based theoretical framework rigorously justifies that the training-free routing mechanism can reliably select relevant experts, amplify their contributions, and yield a robust merged model without additional training overhead.

# C  EXPERIMENT DETAILS

## C.1  EMPLOYED DATASETS AND ASSOCIATED LICENCES

**Discriminative Tasks.**

- **MRPC**. A binary paraphrase detection task from the Microsoft Research Paraphrase Corpus. Each example consists of a pair of sentences, and the model must determine if they are semantically equivalent. It has 3,668 training examples, 408 validation examples, and 1,725 test examples.

- **QQP**. A paraphrase detection task on Quora Question Pairs. The model must decide whether two questions are semantically identical. The training set contains 363,846 examples, with 40,430 for validation, and 390,965 for testing (test labels are not publicly available).

- **MNLI**. A natural language inference (NLI) task with three labels: entailment, neutral, and contradiction. The dataset includes multiple genres of text. It contains 392,702 training examples, 9,815 matched validation, 9,832 mismatched validation, and 20,000 test examples.

- **QNLI**. A binary classification task converted from the Stanford Question Answering Dataset. The model determines whether a given context sentence contains the answer to a question. It consists of 104,743 training examples, 5,463 validation examples, and 5,463 test examples.

- **RTE**. A binary entailment task combining data from multiple RTE challenges. The task is to determine if a hypothesis sentence can be inferred from a given premise. The dataset contains 2,490 training examples, 277 validation examples, and 3,000 test examples.

The licenses of QNLI are licensed under CC-BY-SA. QQP is licensed under MIT. MRPC are licensed under Apache 2.0. MNLI is licensed under OANC. RTE is licensed under CC BY 4.0. Thus, these datasets in GLUE are available for non-commercial research purposes.

**Generation and Math Tasks.** We also incorporate a dataset designed for generative tasks, specifically targeting mathematical reasoning. The **MAWPS** dataset consists of 1,772 examples of math word problems, requiring models to generate the correct mathematical expressions or answers based on natural language descriptions.

**Vision Tasks.**

- **MNIST**. A benchmark dataset for image classification, containing grayscale images of handwritten digits across 10 classes. The training set has 60,000 images, and the test set has 10,000 images, with a balanced distribution among classes.

- **EuroSAT**. A satellite image classification dataset consisting of 27,000 labeled and geo-referenced images across 10 classes.

- **CIFAR-10**. A benchmark for object recognition tasks in computer vision. It consists of 60,000 32x32 color images in 10 different classes, with 6,000 images per class. The dataset is divided into 50,000 training images and 10,000 test images.

- **CarBrands50**. A car classification dataset comprising 50 classes. The dataset contains a total of 4,500 labeled images, which are partitioned into 4,400 images for training, and 100 for validation.

- **FRUITS100**. A fruit classification dataset comprising 100 classes. The dataset contains a total of 50,000 labeled images, which are partitioned into 40,000 images for training, 5,000 for validation, and 5,000 for test.

- **GTSRB**. A traffic sign classification dataset containing over 50,000 images across 43 classes of traffic signs.

- **DTD**. A texture classification dataset with 47 classes and a total of 5,640 images, with approximately 120 images per class.

- **RESISC45**. A remote sensing image scene classification dataset with 45 classes and 31,500 images, approximately 700 per class.

- **GRABAGE**. A grabage classification dataset. The dataset contains a total of 147,674 labeled images, which are partitioned into 133,038 images for training, and 14,642 for test.

- **PLANTS**. A plant classification dataset comprising 30 classes. The dataset contains a total of 30,000 labeled images, which are partitioned into 24,000 images for training, 3,000 for validation, and 3,000 for testing.

## C.2 COMPARATIVE EVALUATION DETAILS

**Funetune Model.** It means that each task uses the corresponding fine-tuned model, which has no interference between tasks but cannot perform multiple tasks simultaneously. It serves as the upper-bound performance for each specific task.

**Multi-task Model.** involving mixing datasets from multiple tasks and training the model jointly, representing one of the earliest solutions for multitask learning.

**Merging Model.** This term denotes algorithms aimed at combining multiple models into a unified, consolidated model, including approaches exemplified by methods Weight Averaging, Task-Arithmetic, Twin-Merging, TR-merging and more.

## C.3 LARGE-MODEL SCALABILITY AND OUT-OF-DOMAIN GENERALIZABILITY DETAILS

As shown in Tables 4, we apply our model merging approach to larger models(Qwen2.5-7B-Instruct), demonstrating that our method scales effectively to models of increased size. Furthermore, as shown in Tables 6, we evaluate the merged models on out-of-domain MMLU benchmark tasks, providing evidence that our approach exhibits strong generalization capabilities beyond the training domains.

Table 6: Detailed Performance on Out-of-Domain MMLU Benchmark Tasks

| MMLU Task | Qwen2.5-7B-Instruct | Weight Averaging | Task-Arithmetic | TR-merging/ours |
|---|---|---|---|---|
| management | 78.6% | 79.6% | 75.7% | 80.2% |
| high_school_world_history | 78.9% | 78.9% | 75.5% | 82.0% |
| college_mathematics | 45.0% | 46.0% | 36.0% | 42.6% |
| high_school_us_history | 79.4% | 80.4% | 77.0% | 79.5% |
| sociology | 82.1% | 82.1% | 77.1% | 83.2% |
| astronomy | 77.0% | 74.3% | 69.1% | 76.9% |
| moral_disputes | 67.1% | 69.4% | 65.9% | 66.2% |
| high_school_government_and_politics | 89.1% | 88.6% | 84.5% | 89.2% |
| medical_genetics | 73.0% | 71.0% | 67.0% | 73.6% |
| high_school_macroeconomics | 72.1% | 72.8% | 70.0% | 73.2% |
| international_law | 76.9% | 78.5% | 72.7% | 77.5% |
| high_school_geography | 83.3% | 83.8% | 80.3% | 84.4% |
| electrical_engineering | 63.4% | 60.0% | 57.9% | 63.5% |
| virology | 48.8% | 50.0% | 47.6% | 48.2% |
| high_school_european_history | 74.5% | 76.4% | 70.3% | 77.0% |
| elementary_mathematics | 60.6% | 62.2% | 56.3% | 62.0% |
| moral_scenarios | 22.5% | 20.1% | 22.0% | 23.0% |
| formal_logic | 50.8% | 49.2% | 42.1% | 52.2% |
| machine_learning | 40.2% | 44.6% | 46.4% | 42.6% |
| us_foreign_policy | 86.0% | 85.0% | 82.0% | 85.6% |
| high_school_psychology | 85.7% | 85.1% | 81.5% | 86.3% |
| high_school_chemistry | 61.6% | 58.1% | 55.7% | 63.2% |
| computer_security | 78.0% | 76.0% | 72.0% | 77.6% |
| college_physics | 53.9% | 54.9% | 50.0% | 54.5% |
| professional_law | 45.8% | 43.9% | 38.9% | 46.5% |
| marketing | 89.7% | 88.5% | 85.9% | 90.3% |
| prehistory | 76.5% | 76.9% | 74.1% | 76.5% |
| college_biology | 80.6% | 83.3% | 78.5% | 80.5% |
| nutrition | 70.6% | 71.2% | 65.4% | 72.2% |
| professional_medicine | 78.7% | 76.8% | 74.6% | 77.4% |
| human_sexuality | 75.6% | 69.5% | 64.9% | 75.4% |
| philosophy | 67.2% | 69.1% | 63.3% | 67.8% |
| high_school_statistics | 71.8% | 71.8% | 66.7% | 70.0% |
| business_ethics | 68.0% | 72.0% | 68.0% | 68.6% |
| professional_accounting | 54.3% | 52.8% | 52.8% | 57.1% |
| high_school_mathematics | 45.6% | 43.0% | 43.0% | 47.3% |
| global_facts | 40.0% | 32.0% | 36.0% | 39.6% |
| miscellaneous | 81.4% | 81.7% | 78.5% | 81.7% |
| anatomy | 71.1% | 70.4% | 70.4% | 72.5% |
| security_studies | 67.8% | 69.0% | 64.5% | 69.6% |
| public_relations | 67.3% | 65.5% | 60.9% | 67.0% |
| clinical_knowledge | 76.6% | 73.6% | 72.8% | 77.4% |
| high_school_physics | 57.0% | 52.3% | 48.3% | 58.2% |
| econometrics | 56.1% | 54.4% | 51.8% | 59.4% |
| conceptual_physics | 69.4% | 70.6% | 66.4% | 69.5% |
| high_school_computer_science | 78.0% | 76.0% | 70.0% | 78.6% |
| college_chemistry | 47.0% | 46.0% | 49.0% | 46.6% |
| high_school_biology | 81.6% | 81.9% | 78.4% | 82.2% |
| world_religions | 83.0% | 79.5% | 75.4% | 84.2% |
| human_aging | 69.1% | 69.1% | 65.9% | 70.6% |
| college_medicine | 68.2% | 70.5% | 64.7% | 70.5% |
| college_computer_science | 60.0% | 54.0% | 47.0% | 62.6% |
| jurisprudence | 75.9% | 75.0% | 71.3% | 76.5% |
| high_school_microeconomics | 81.1% | 84.5% | 74.8% | 82.1% |
| abstract_algebra | 49.0% | 48.0% | 38.0% | 51.6% |
| professional_psychology | 70.4% | 70.9% | 64.4% | 71.0% |
| **Avg** | 67.9% | 67.3% | 63.6% | **67.5%** |

