# OpenReview forum: "TR-MERGING: TRAINING-FREE ROUTER FOR MODEL MERGING"
_ICLR.cc/2026/Conference — Submitted to ICLR 2026_

### Official Review · Reviewer_ykfp · 2025-10-24

**Soundness:** 2
**Presentation:** 2
**Contribution:** 2
**Rating:** 2
**Confidence:** 4

**Summary:**

In this paper, the authors propose TR-Merging, a training-free router for model merging that dynamically selects and weights expert models based on semantic similarity between input data and task domains. The method leverages a pre-trained embedding model to compute cosine similarities, applies temperature scaling and top-k selection to enhance discriminability, and merges parameters without any additional training. The authors validate their approach across a range of NLP and CV tasks, demonstrating competitive or superior performance compared to training-based router methods, while maintaining computational efficiency and scalability.

**Strengths:**

- The paper proposes a conceptually simple yet effective training-free routing strategy. It leverages LoRA to reduce memory overhead and avoids the often-overlooked costs and complexities associated with training additional routing networks.
- This method has validated its effectiveness across multiple domains (NLP, CV) and presented robust empirical results.
- The paper provides solid theoretical analysis, which enhances the credibility of the method.

**Weaknesses:**

- The experimental details are considerably inadequate. Specifically, it fails to provide the specific values of temperature ($\tau$) and sample size ($M$) for each domain, nor does it include corresponding ablation experiments to illustrate the effectiveness.
- Following the approach outlined in the paper, if we directly select the model with the smallest similarity as the merged model—i.e., setting the coefficients of all other models to 0—could this potentially yield better performance?
- The discussion on the specific advantages of LoRA is rather brief, and there is a lack of ablation experiments comparing full-parameter merging with LoRA-based merging. This is particularly notable given that all baselines employed in the paper utilize full-parameter settings as reported in their original sources.
- There are issues with certain figures and textual descriptions:
    - The captions for Figure 1, 2, 3 and Table 1, 2, 3, 4, 5 in the main text fail to provide useful information.
    - There are some typos. For instance, in the "Implementation detail" section (Section 4.1), NLP tasks are incorrectly described as being trained using ViT.

**Questions:**

My questions are listed with my weaknesses above.

I'm excited to engage with the authors to clear up the aspects I don't fully understand and I'm optimistic that with some iteration this paper can be made stronger.

---

### Official Review · Reviewer_v3ys · 2025-10-30

**Soundness:** 2
**Presentation:** 1
**Contribution:** 1
**Rating:** 2
**Confidence:** 5

**Summary:**

While existing high-performance model merging methods often utilize a router to dynamically select experts based on the input, this router typically requires an additional training phase, incurring further data and computational overhead. This paper proposes a training-free router based on embedding similarity to resolve this issue.

**Strengths:**

The paper correctly identifies and highlights the dependency on an additional training phase as a significant limitation of existing router-based model merging methodologies.

**Weaknesses:**

1. Although the router itself is not trained, the method still necessitates access to the training data to compute domain similarity. This dependency remains a significant limitation. Validation is needed on how this method performs if it cannot access data from the original training distribution.
2. The method's performance appears to be heavily dependent on the chosen pre-trained embedding model (the "router"). This dependency is likely to be a critical factor when the types of datasets or domains change.
3. Furthermore, the process of generating these embeddings via a powerful pre-trained model (like CLIP) could introduce significant inference latency. This may result in the method being slower than existing approaches that use a lightweight, trained router, contrary to the paper's claims.
4. The paper lacks sufficient analysis (e.g., ablation studies) of key hyperparameters that could significantly affect performance. This includes the sample size (M) used to represent each task domain and the temperature coefficient.
5. The core technical components, such as temperature scaling and softmax normalization, appear to be standard, existing methods applied directly, rather than novel contributions.
6. The number of compared SOTA merging techniques is insufficient. The comparison would be much stronger if it included other recent router-based merging methods.
7. The overall readability and presentation of the paper are poor. The placement and formatting of figures and tables, as well as the prose, require significant revision for clarity.

**Questions:**

1. Twin-Merging reportedly uses a very lightweight MLP-based router. How can TR-Merging, which uses a much heavier model like CLIP as its router, achieve faster inference times as reported in the tables? This result seems counter-intuitive and requires a detailed explanation of the experimental setup for measuring latency.
2. It seems that the choice of the pre-trained model used as the router (e.g., CLIP vs. BGE vs. other alternatives) would have a substantial impact on performance. Could the authors comment on this dependency or provide analysis on using different embedding models as the router?

---

### Official Review · Reviewer_DovB · 2025-10-31

**Soundness:** 2
**Presentation:** 3
**Contribution:** 2
**Rating:** 4
**Confidence:** 3

**Summary:**

The paper proposes TR-Merging, a method for merging multiple fine-tuned models by using a dynamic, input-dependent weighting scheme. The core idea is to bypass the need for an explicitly trained routing module, which is a key component in recent high-performance methods like Twin-Merging. Instead, TR-Merging employs a "training-free router" that calculates the cosine similarity between a new input's embedding and the pre-computed average embeddings of small data samples from each expert model's training domain. These similarity scores determine the weights for merging the expert models' parameters for the current inference task. The authors present experiments across NLP and CV tasks, arguing that their method achieves competitive performance with router-based approaches while being more efficient and simpler to implement.

**Strengths:**

The central concept of using semantic similarity to guide the merging process is simple, elegant, and easy to understand. It provides a clever alternative to training a dedicated routing network.

The method shows competitive performance on several benchmarks when compared to established baselines, including a state-of-the-art method that requires router training. This suggests the direction has potential.

Compared to methods requiring training a router, the proposed approach is conceptually simpler and avoids a complex training pipeline, provided a suitable embedding model is available.

**Weaknesses:**

The paper's most significant flaw is the complete omission of crucial hyperparameter values (M, τ, K) and the methodology for their selection. Without this information, the experiments are not reproducible, and the validity of the results cannot be independently verified. It is impossible to know if the performance is robust or the result of careful, undisclosed tuning.

The core marketing of the method as "training-free" is questionable. The method's success is entirely dependent on a large, pre-trained embedding model. This dependency is not analyzed. The paper fails to investigate how performance changes with different or weaker embedding models. This makes the efficiency claims (e.g., "no training cost") incomplete, as the cost is simply outsourced.

The paper lacks a proper ablation study on its key components. What is the sensitivity to M, the number of samples? How does the choice of embedding model affect performance? Answering these questions is fundamental to understanding the method, not just demonstrating it works on one specific setup.

The claim of successfully merging generative and discriminative tasks is based on extremely thin evidence. The reported gains are marginal (+0.5%), and the experimental setup and baseline are not sufficiently detailed. This feels like a speculative claim rather than a demonstrated capability.

**Questions:**

N/A

---

### Official Review · Reviewer_eJCb · 2025-11-03

**Soundness:** 3
**Presentation:** 3
**Contribution:** 3
**Rating:** 6
**Confidence:** 2

**Summary:**

The paper proposes TR-Merging, a training-free router framework for model merging that integrates multiple fine-tuned models (experts) without additional training.

**Strengths:**

1. The introduction of a training-free router is a simple yet impactful idea, addressing a key limitation of router-based merging (training cost).
2. Evaluation across both CV and NLP domains, with comparisons to strong baselines (Fisher, Task Arithmetic, TIES, Twin-Merging).

**Weaknesses:**

1. The cosine-similarity-based routing is conceptually close to classical nearest-neighbor or retrieval-based gating. The paper would benefit from clearer distinction and analysis versus simple centroid-matching baselines.

**Questions:**

1. How sensitive is TR-Merging to the choice of the embedding model used as the router?
2. What happens if the expert domains are highly overlapping (e.g., MNLI vs. QNLI)?

---

### Meta-Review · Area_Chair_oMoq · 2026-01-03

**Summary:**

This paper proposes TR‑Merging, a training‑free router for model merging based on embedding similarity. While the idea of avoiding router training is appealing, the work suffers from major shortcomings. Critical hyperparameters are missing, making the experiments non‑reproducible. The method’s reliance on large pre‑trained embedding models undermines the “training‑free” claim and raises efficiency concerns. Reported gains are marginal, ablation studies are absent, and comparisons to recent router‑based methods are incomplete. Presentation quality is also weak, with unclear figures and typos. Hence, I recommend rejection.

**Reviewer Concerns:**

The authors chose to not post a rebuttal.

**Reviewer Scores:**

The authors chose to not post a rebuttal. Hence, the scores are unchanged.

---

### Decision · Program_Chairs · 2026-01-26

Reject